# An MRF-UNet Product of Experts for Image Segmentation

**Mikael Brudfors**[1]                                      MIKAEL.BRUDFORS@KCL.AC.UK
**Yaël Balbastre**[2]                                        YBALBASTRE@MGH.HARVARD.EDU
**John Ashburner**[3]                                          J.ASHBURNER@UCL.AC.UK
**Geraint Rees**[4]                                                G.REES@UCL.AC.UK
**Parashkev Nachev**[5]                                          P.NACHEV@UCL.AC.UK
**Sébastien Ourselin**[1]                               SEBASTIEN.OURSELIN@KCL.AC.UK
**M. Jorge Cardoso**[1]                                   M.JORGE.CARDOSO@KCL.AC.UK

[1] *School of Biomedical Engineering & Imaging Sciences, KCL, London, UK*

[2] *Athinoula A. Martinos Center for Biomedical Imaging, MGH and HMS, Boston, USA*

[3] *Wellcome Center for Human Neuroimaging, UCL, London, UK*

[4] *Institute of Cognitive Neuroscience, UCL, London, UK*

[5] *Institute of Neurology, UCL, London, UK*

## Abstract

While convolutional neural networks (CNNs) trained by back-propagation have seen unprecedented success at semantic segmentation tasks, they are known to struggle on out-of-distribution data. Markov random fields (MRFs) on the other hand, encode simpler distributions over labels that, although less flexible than UNets, are less prone to over-fitting. In this paper, we propose to fuse both strategies by computing the product of distributions of a UNet and an MRF. As this product is intractable, we solve for an approximate distribution using an iterative mean-field approach. The resulting MRF-UNet is trained jointly by back-propagation. Compared to other works using conditional random fields (CRFs), the MRF has no dependency on the imaging data, which should allow for less over-fitting. We show on 3D neuroimaging data that this novel network improves generalisation to out-of-distribution samples. Furthermore, it allows the overall number of parameters to be reduced while preserving high accuracy. These results suggest that a classic MRF smoothness prior can allow for less over-fitting when principally integrated into a CNN model. Our implementation is available at https://github.com/balbasty/nitorch.

**Keywords:** CNN, U-Net, MRF, products of experts, image segmentation.

## 1. Introduction

This paper concerns the task of semantic image segmentation: labelling each voxel of an image with a corresponding class. Robustly identifying voxels of organs or lesions from medical images is one of the more challenging tasks in medical image analysis. In this domain, magnetic resonance imaging (MRI) is an extremely versatile modality, as a variety of image contrasts can be obtained by changing the multitude of parameters encoding the MR sequence. However, the resulting image is extremely sensitive to both scanner- and subject-specific parameters (*e.g.*, field strength, homogeneity of the different magnetic fields, loading of the coils). This makes building generic segmentation tools – that work on any contrast or resolution – extremely challenging due to the domain shift introduced by both subject and scanner variability.

While classical probabilistic methods, which model the different sources of artefacts and optimise parameters on a subject-wise basis, generally work well on out-of-distribution data (Ashburner and Friston, 2005; Zhang et al., 2001; Fischl et al., 2004; Van Leemput et al., 1999), neural networks (NNs) have so far struggled to generalise in the same way (Dolz et al., 2018). This issue is intrinsically linked to the flexibility of NNs, which makes them extremely good at recognising patterns but also blind to invariances that are not present in the data they were trained on. The first methods that tackled generalisation issues in NN segmentation therefore aimed to pre-process the images to standardise their intensity profiles (Zhuge and Udupa, 2009; Weisenfeld and Warfield, 2004; Han and Fischl, 2007). However, these techniques do not make the networks generalise *per se* but merely remove (some) variance from the data.

More recently, two different paths have been taken to build segmentation networks that are insensitive to certain image characteristics. The first approach relies on data augmentation, with the underlying idea that, for NNs to be invariant to some feature, they need this invariance to be discoverable from the training data. The idea is that the feature (*e.g.*, intensity non-uniformities) can be modelled, and therefore sampled. Spatial augmentation, for example, was quickly adopted to present NNs with many more brain shapes than would be possible using real images alone (Pereira et al., 2016; Castro et al., 2018). As for appearance augmentation, Jog et al. (2019) generated realistic images with a variety of contrasts using a pulse-sequence simulator and used them to train a UNet on multiple contrasts. In Billot et al. (2020a), the idea was pushed further by generating a multitude of MR contrasts from pre-segmented MRIs. Importantly, these simulations did not aim to generate realistic images, but to build contrast invariance in the training set. More work has since been extended to build invariance to resolution (Billot et al., 2020b), or even to image features entirely (Hoffmann et al., 2020). When labelled images are scarce, augmentation can be used in conjunction with a consistency loss in a semi-supervised setting to enforce consistency between predictions obtained from the same original images augmented in different ways (Xie et al., 2019). It has also recently been proposed to add spatial regularization, such as total variation, to the segmented object and solve by gradient decent (Jia et al., 2021). The second approach focuses on the architecture of the NNs, such that invariance is directly built-in, independent of the training data. For example, by adding a new set of batch normalisation parameters in the network as it encounters training data from a new acquisition protocol (Karani et al., 2018). Adversarial techniques can also be used to learn feature representations agnostic to the data domain. This can be achieved by learning an adversarial network that attempts to discriminate the domain of the input data coming from both domains (Kamnitsas et al., 2017a). Finally, transfer learning is one more popular method for improving the generalisability of NNs (Knoll et al., 2019).

A modelling paradigm that can be used to introduce both augmentation and architectural components is based on probability theory (Jaynes, 2003). In probabilistic models, the joint distribution over all variables (observed and hidden) is factorised in a way that reveals components that influence each observed sample, and components that embed general knowledge, independent of a particular sample. These components are commonly denoted as likelihood and prior, respectively. Conversely, most NNs compute a function that map observed data and hence cannot separate prior and data components. One line of research aims to bridge the gap between classical probabilistic models and NNs. In Brudfors et al.

(2019), it was shown that inference under a low parameter Markov random field (MRF) prior could be formalised as a feed-forward NN. The concept was used to encode complex non-linear MRFs that cannot be optimised in a classical maximum-likelihood framework, but can be optimised by back-propagation. This MRF was then used to simply post-process segmentations obtained from a probabilistic segmentation model.

In the present work, this idea is extended so that a UNet is used in place of the probabilistic model. In this context, the UNet and MRF are considered as independent segmentation experts (in the sense that they define probability distributions over possible segmentations), whose beliefs should be merged in order to take an informed and balanced decision. Here, this 'belief fusion' is performed by taking the product of their distributions (Hinton, 2002). As normalising this product is intractable, we use variational inference to estimate the closest factorised distribution under the Kullback-Leibler divergence. As in Brudfors et al. (2019), we find that this mean-field inference can be formalised as a recurrent NN that is appended to the UNet. Finally, we propose to jointly train the UNet and MRF by back-propagation. Encoding the MRF in the NN has advantages over other works that use CRFs (Zheng et al., 2015; Chen et al., 2015; Kamnitsas et al., 2017b; Monteiro et al., 2018), as the MRF is a prior distribution over the segmentation labels, rather than a conditional distribution. This property allows the MRF to regularise the segmentation labels alone, without being influenced by the image data, which could allow for less over-fitting. Additionally, the performance of CRFs have been shown to not translate well to the medical imaging domain (Monteiro et al., 2018). Conversely, we show in this paper that the MRF-UNet improves segmentation accuracy on both in- and out-of-distribution 3D brain MRIs. We also show that it allows the overall number of CNN parameters to be reduced with higher accuracy preserved. These results suggest that combining a classical type of MRF prior with a highly parametrised segmentation CNN could improve segmentation accuracy and generalisability.

## 2. Methods

Let us consider the segmentation problem where an observed intensity image $\mathbf{X} \in \mathbb{R}^{I \times C}$, with $I$ voxels and $C$ channels, is segmented into $K$ classes. The segmentation can be encoded in a one-hot label image $\mathbf{Z} \in \{0, 1\}^{I \times K}$. In the supervised setting, a set of training pairs $\{\mathbf{X}_n, \hat{\mathbf{Z}}_n\}_{n=1}^N$ is available. This set is used to optimise a function $\mathcal{F}(\mathbf{X})$ that predicts a segment from an image. Currently, functions of choice are convolutional neural networks (CNNs); often some flavour of UNet (Long et al., 2015; Ronneberger et al., 2015). The CNN parameters are found by back-propagating gradients from an appropriate loss function.

Segmentation UNets typically end with a softmax activation function, which ensures that their output, $\boldsymbol{\pi} \in [0, 1]^{I \times K}$, can be interpreted as probabilities. We can therefore see the network as encoding a product of (posterior) categorical distributions:

$$p(\mathbf{Z} \mid \mathbf{X}, \mathcal{F}) = \prod_{i=1}^{I} \mathrm{Cat}\left(\mathbf{z}_i \mid \mathcal{F}_i(\mathbf{X})\right) = \prod_{i=1}^{I} \prod_{k=1}^{K} \mathcal{F}_{ik}(\mathbf{X})^{z_{ik}} . \tag{1}$$

Here, $\mathcal{F}$ denotes the UNet parameters and $\mathcal{F}(\mathbf{X})$ the result of its forward pass. The subscripts $i$ and $k$ respectively denote extracting a single voxel and a single class.

On the other hand, an MRF is a joint probability over all voxels, with the property that the conditional probability of a voxel, given all others, only depends on a small neighbour-

---

**Algorithm 1:** MRF-UNet forward pass.

---

**Input: X**, **R**     (image data, initial responsibilities)
**Output: R$^\star$**     (VB optimal responsibilities)
$\mathbf{U} \leftarrow \text{UNet}(\mathbf{X}; \mathcal{F})$;
$\mathbf{U} \leftarrow \mathbf{U} - \text{log-sum-exp}(\mathbf{U})$;
$\mathbf{R} \leftarrow 1/K$;
**for** $i \leftarrow 1$ **to** $niter$ **do**
  |   $\mathbf{R} \leftarrow \text{softmax}\left(\mathbf{U} + \text{MRF}(\mathbf{R}; \mathcal{W})\right)$;     (Eq. (7))
**end**

---

hood $\mathcal{N}$:

$$p\left(\mathbf{z}_i \mid \{\mathbf{z}_j\}_{j \neq i}, \mathcal{W}\right) = p\left(\mathbf{z}_i \mid \mathbf{z}_{\mathcal{N}_i}, \mathcal{W}\right) \; , \tag{2}$$

where $\mathcal{W}$ denotes the MRF weights. We make the assumption that this neighbourhood is stationary, meaning that it is defined by relative positions with respect to $i$. We additionally assume that it factorises over its neighbours and that each factor is a categorical distribution:

$$p(\mathbf{z}_i \mid \mathbf{z}_{\mathcal{N}_i}, \mathcal{W}) = \prod_{\delta \in \mathcal{N}_i} \prod_{k=1}^{K} \prod_{l=1}^{K} (w_{kl,\delta})^{z_{ik} \cdot z_{i+\delta,l}} \; . \tag{3}$$

This paper uses a first-order neighbourhood, but a larger one could also have been used.

The UNet and MRF distributions can be fused by taking their product and normalising (Hinton, 2002):

$$p\left(\mathbf{Z} \mid \mathbf{X}, \mathcal{F}, \mathcal{W}\right) = \frac{p\left(\mathbf{Z} \mid \mathcal{W}\right) p\left(\mathbf{Z} \mid \mathbf{X}, \mathcal{F}\right)}{\int_{\mathbf{Z}} p\left(\mathbf{Z} \mid \mathcal{W}\right) p\left(\mathbf{Z} \mid \mathbf{X}, \mathcal{F}\right) \mathrm{d}\mathbf{Z}} \; . \tag{4}$$

However, the conditional distribution on the left-hand side is clearly intractable. Instead, we make a mean-field approximation and look for an approximate distribution $q(\mathbf{Z}) = \prod_i q_i(\boldsymbol{z}_i)$, which factorises across voxels and is closest to the true product of distributions in terms of their Kullback-Leibler divergence $\text{KL}\left(q\|p\right)$. As in variational Bayesian inference, we can update the distribution of a factor by taking the expected value of the true product of distributions with respect to all the others factors (Bishop, 2006):

$$\ln q^\star(\mathbf{z}_i) = \mathbb{E}_{q_{j \neq i}}\left[\ln p\left(\mathbf{Z} \mid \mathbf{X}, \mathcal{F}, \mathcal{W}\right)\right] + \text{const} \tag{5}$$

$$= \sum_{k=1}^{K} z_{ik} \left(\ln \mathcal{F}_{ik}(\mathbf{X}) + \sum_{\delta \in \mathcal{N}} \sum_{l=1}^{K} \mathbb{E}_q\left[z_{i+\delta,l}\right] \ln w_{kl,\delta}\right) + \text{const} \; . \tag{6}$$

Let us write $\mathbb{E}_q\left[\mathbf{z}_j\right] = \mathbf{r}_j$. We note that the second term can be seen as the convolution of the map $\mathbf{R}$ with a small kernel whose weights are $\ln w_{kl\delta}$ and center weight is zero (Brudfors et al., 2019). We denote this convolution $\mathcal{W} * \mathbf{R}$ and we recognise that $q_i^\star$ is a categorical distribution $\text{Cat}(\mathbf{z}_i \mid \mathbf{r}_i^\star)$ with parameter:

$$\mathbf{r}_i^\star = \text{softmax}\left(\ln \mathcal{F}_i(\mathbf{X}) + [\mathcal{W} * \mathbf{R}]_i\right) . \tag{7}$$

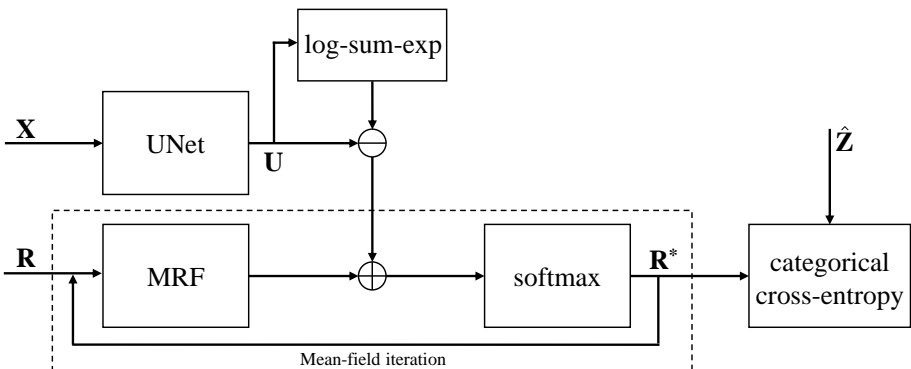

Figure 1: Schematic illustration of the MRF-UNet product. An image $\mathbf{X}$ is passed forward, through a UNet, whose logit outputs are then fused with the current estimate of the responsibility map $\mathbf{R}$. The responsibility map is updated in an iterative fashion $\mathbf{R}^{\star}$. For training, the categorical cross entropy between the reference segmentation $\hat{\mathbf{Z}}$ and the responsibilities is computed.

Note that, by letting the UNet output logits maps (pre-softmax), $\ln \mathcal{F}_i(\mathbf{X})$ can be formulated using the log-sum-exp trick[1]. The expression in (7) gives us the optimal expected label image, as for the categorical distribution we have $\mathbb{E}_{q^*}[z_{ik}] = r_{ik}^{\star}$.

In a variational setting, optimising for a segmentation $\mathbf{R}^{\star}$ involves iterating over the expression in (7), which minimises the KL divergence. To discourage the CNN from overfitting to a fixed number of iterations we randomly sample the number of iterations from a discrete uniform distribution during training, but keep this number fixed during testing. The MRF-UNet forward pass is described in Algorithm 1 and also visualised in Figure 1.

## 3. Validation

In this section we compare the proposed MRF-UNet architecture to a baseline model (*i.e.*, a MRF-UNet without the MRF component), for segmenting publicly available 3D MRI brain scans. We compare the segmentation accuracy of the two methods on in- and out-of-distribution test data, and how it depends on the number of network parameters. We additionally investigate the iterative nature of the MRF-UNet.

### 3.1. Data

The following MR images from two publicly available datasets are used:

- **MICCAI2012**[2]: T1-weighted MRIs of 30 healthy subjects aged 18 to 96 years, (mean: 34, median: 25). The scans were manually segmented into 136 anatomical regions (by Neuromorphometrics Inc.) for the MICCAI 2012 multi-atlas segmentation

---

1. $\log(\mathrm{softmax}(\mathbf{x}))_k = x_k - \log(\sum_j^K \exp(x_j)) = x_k - (x^{\star} + \log\left[\sum_j^K \exp\left(x_j - x^{\star}\right)\right])$, where $x^{\star} = \max(\mathbf{x})$.

2. https://my.vanderbilt.edu/masi/workshops/

challenge. We combined regions to form four labels: gray matter (GM), white matter (WM), ventricles (VEN) and other (OTH).

- **MRBrainS18**[3]: T1-weighted MRIs of seven subjects all aged 50 years or older (some with pathology). The scans were manually segmented into ten anatomical regions by the same neuroanatomist. From these regions we selected the GM, WM, VEN and OTH labels, for parity with MICCAI2012.

Within each dataset, all subjects were imaged on the same scanner and with the same sequences, whilst between datasets, the scanners and sequences differ. An example subject, from both datasets, is shown in Figure 2.

## 3.2. Implementation

The UNet has five encoding/decoding layers and use convolutional filters with $3 \times 3 \times 3$ kernels and stride of two. In our experiments, we vary the number of filters for the encoding layers are (which are 'mirrored' for the decoding layer). These layers are followed by a final convolution layer that outputs $K$ channels. The MRF network has an initial $3 \times 3 \times 3$ MRF layer, whose centre weights are zero with $K^2$ filters, this is followed by one $1 \times 1 \times 1$ convolution with $K$ filters. The baseline UNet ends with a softmax activation function, whereas there is no final activation in the UNet component (nor the MRF component) of the MRF-UNet product. Instead, their logits are summed before being softmaxed, as depicted in Figure 1. All layers in both networks use leaky ReLU activations ($\alpha = 0.2$). The networks are optimised using categorical cross-entropy and the ADAM optimiser (lr=$10^{-3}$), where the learning rate is dynamically reduced based on the difference in subsequent values of the validation loss. During training, we augment with random diffeomorphic deformations, multiplicative smooth intensity non-uniformities, and additive Gaussian noise. The MRF-UNet uses $n_{\mathrm{iter}} = 10$ mean-field iterations during testing and $n_{\mathrm{iter}} \sim \mathcal{U}\{5, 15\}$ during training. We train for a fixed number of 50 epochs, with a batch size of one. Our implementation was done using PyTorch.

## 3.3. Experiments

The MICCAI2012 dataset is used as in-distribution data with a (train, validation, test) split of (13, 3, 14). All seven MRBrainS18 images is used as out-of-distribution data and considered solely for testing. Both the UNet and the MRF-UNet are trained on the MICCAI2012 training set; then, mean pairwise Dice scores are computed for predicting the GM, WM, VEN and OTH labels on the MICCAI2012, as well as the MRBrainS18, test subjects. This is then done for a varying number of UNet parameters: the number of filters in the encoding and decoding layers are set to ($2^j$, $2^{(j+1)}$, $2^{(j+2)}$, $2^{(j+3)}$, $2^{(j+4)}$) for $j = \{1, 2, 3, 4, 5, 6\}$ (flipped for the decoding layer). We also perform a simple convergence analysis, where a trained MRF-UNet model ($j = 5$) is fitted to the in- and out-of-distribution data, varying the number of mean-field iterations from 0 to 20, and computing the average Dice score.

---

3. https://mrbrains18.isi.uu.nl/

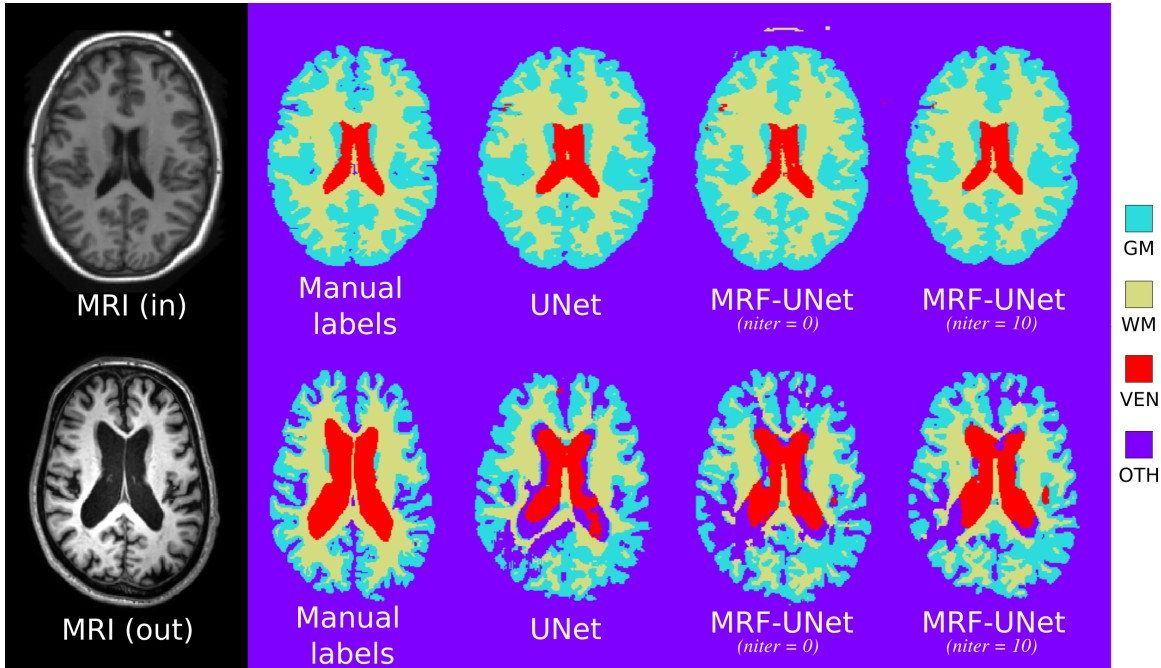

Figure 2: Random example segmentation results for in- (top) and out-of-distribution (bottom) data, as axial slices. The MRI, with its ground-truth, manual label image, has been segmented either with a UNet or an MRF-UNet ($j = 3$). For the MRF-UNet we show results for zero (without applying the MRF) and 10 mean-field iterations. A reason that the MRF-UNet ($niter = 0$) segmentation differs from the UNet could be that training a combined MRF-UNet allows the UNet part to focus on certain image features, as the MRF models some of the spatial regularity. Therefore, when one does a forward pass though a trained MRF-UNet, without any mean-field iterations, some of the spatial smoothness is missing in the segmentation.

### 3.4. Results

Figure 3 shows the resulting Dice scores from the in- and out-of-distribution segmentation tasks. The median Dice scores across labels, for each parameter configuration, were for the in-distribution task (UNet+MRF vs. UNet): 0.84 vs 0.82 for $j = 1$, 0.90 vs 0.86 for $j = 2$, 0.92 vs 0.91 for $j = 3$, 0.92 vs 0.90 for $j = 4$, 0.92 vs 0.92 for $j = 5$, 0.91 vs 0.87 for $j = 6$; and for the out-of-distribution task: 0.73 vs 0.67 for $j = 1$, 0.80 vs 0.67 for $j = 2$, 0.83 vs 0.79 for $j = 3$, 0.81 vs 0.79 for $j = 4$, 0.84 vs 0.83 for $j = 5$, 0.84 vs 0.81 for $j = 6$. Paired Wilcoxon tests with Bonferroni correction show that the segmentation results for $j = \{1, 2, 3, 4, 5, 6\}$ are significant, except for $j = 5$, for both datasets. That is, the MRF-UNet outperforms the baseline UNet for almost all parameter configurations. Furthermore, the plot implies that the MRF-UNet model allows for using fewer UNet parameters, with retained Dice scores. Introducing the MRF adds parameters to the MRF-UNet, which could results in a better fit;

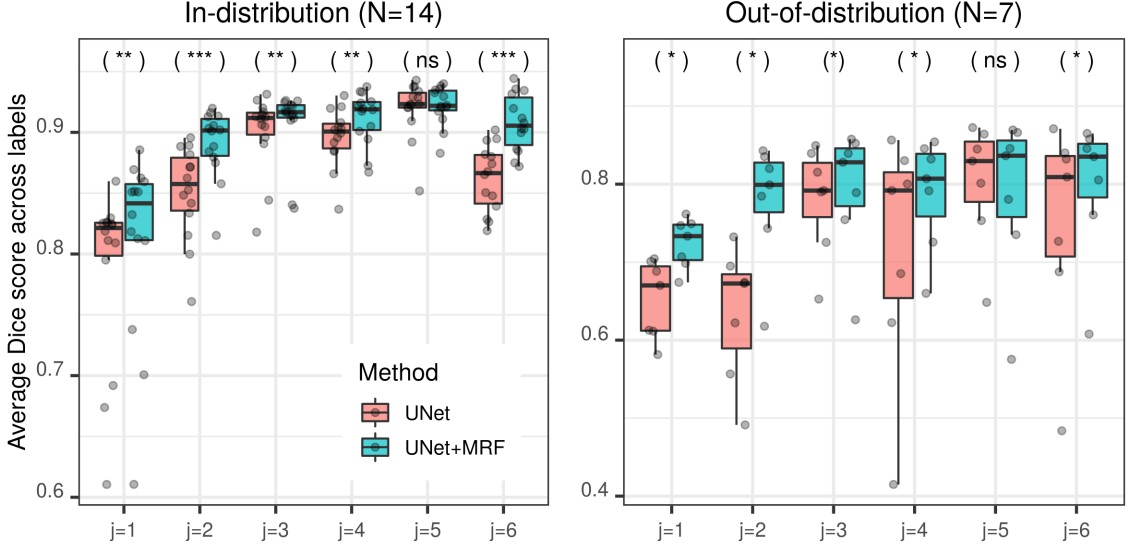

Figure 3: Average Dice scores across labels for segmenting the in- and out-of-distribution test images into GM, WM, VEN and OTH; using both the UNet and the MRF-UNet. For both networks, we vary the number of convolutional filters: $(2^j, 2^{(j+1)}, 2^{(j+2)}, 2^{(j+3)}, 2^{(j+4)})$, for $j = \{1, 2, 3, 4, 5, 6\}$. On each box, the central mark indicates the median, and the bottom and top edges of the box indicate the 25th and 75th percentiles, respectively. The whiskers extend to the most extreme data points not considered outliers. The asterisks above the boxes indicate statistical significance of paired Wilcoxon tests after Bonferroni correction: 0.05 $(*)$, 0.01 $(**)$, 0.001 $(* * *)$.

however, even for the smallest architecture considered in our experiments, (2, 4, 8, 16, 32), the increase in parameters is less than 2.5%. For the largest architecture, this drops to less than 0.0001%, which shows how lightweight the MRF component is. Figure 2 shows example segmentations for both datasets. Segmenting out-of-distribution images is clearly a very challenging task, having only seen the in-distribution data. However, it can be seen, from comparing the MRF-UNet with 0 mean-field iterations (no MRF applied) to 10 iterations, that the MRF component behaves as expected, encouraging neighbouring voxels to have similar labels. Figure 4 shows the results of the convergence analysis. For both the in- and out-of-distribution data, the validation Dice converges quickly and monotonically. The analysis suggests that no more than ten iterations may be needed, which is beneficial from both a memory and a runtime point-of-view. It is furthermore encouraging that the learned mean-field iterative approach replicates the monotonically increasing nature of variational updates.

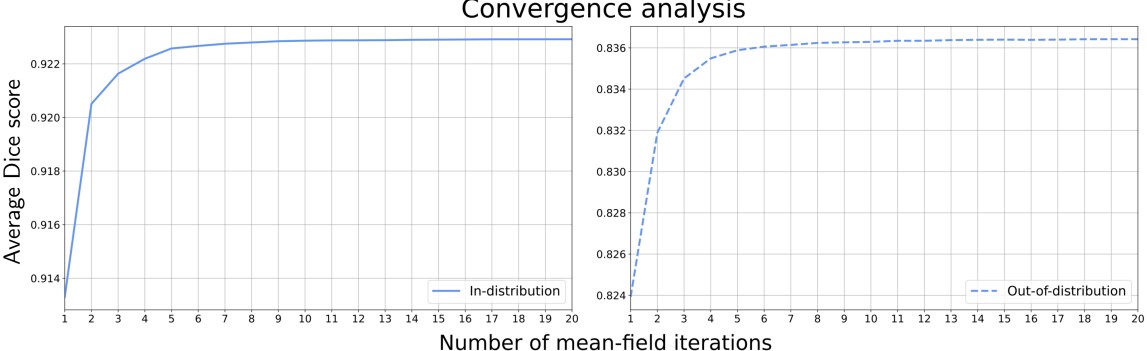

Figure 4: Convergence analysis of the learned mean-field iterative approach, on the in- (left) and out-of-distribution (right) test images. Average Dice scores were computed for an increasing number of iterations.

## 4. Conclusion

In this paper, we described a novel approach for combining a segmentation CNN with a low parameter, first-order MRF prior over the image labels. Our hypothesis was that this 'simple' prior would learn abstract, label-specific features and thereby improve segmentation accuracy on both in- and out-of-distribution data. We showed the validity of this assumption on 3D MR images of the human brain. Future work will extend this validation to data from other domains. One could argue that explicitly encoding prior information into a high-dimensional model, such as a CNN, is superfluous, as the CNN should implicitly capture this information from training data. However, in the interest of limited data, and model complexity, explicit priors still play an important role.

Readers familiar with unsupervised segmentation techniques may notice that the expression for updating the prediction of a segmentation in (7), coincides with updating the expected posterior over latent segmentation labels, where the likelihood is a mixture model and the prior an MRF (Langan et al., 1992; Van Leemput et al., 1999). In this work, it is not possible to encode the posterior using Bayes' rule as the UNet outputs a conditional distribution over segmentation labels, not imaging data; and we here chose to instead use a products of experts model. However, the connection between the two methods is clear and could inspire future extensions of our approach.

## Acknowledgments

MB, PN and MJC were supported by Wellcome Innovations [WT213038/Z/18/Z]. PN was supported by the UCLH NIHR Biomedical Research Centre. YB was supported by the National Institutes of Health under award numbers U01MH117023, R01AG064027 and P41EB030006.

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
