# OpenReview forum: "An MRF-UNet Product of Experts for Image Segmentation"
_MIDL.io/2021/Conference — MIDL 2021_

### Official Review · AnonReviewer3 · 2021-03-07

**Confidence:** 4
**Preliminary Rating:** 2
**Final Rating:** 3

**Summary:**

This work proposes the MRF-Unet, which fuses an MRF smoothness prior into a Unet model for image segmentation. The paper extends the prior work by Brudfors et al., IPMI 2019, which posed the MRF prior as a feed-forward neural network. The product of the MRF and Unet distributions are approximated using an iterative mean-field approach, and the resulting model can be trained using standard backpropagation. Using 2 brain tissue segmentation datasets, they demonstrated improved accuracy on both in and out of distribution tests compared to the basic Unet alone.

**Strengths:**

1. This work incorporates a label smoothness prior into a neural network based segmentation model, which can all be optimized via backpropagation. This ability to include such prior smoothness information into the neural network, which usually only looks to optimize the loss of pixels independently, should improve the quality of the image segmentation.

2. The paper is fairly well-organized, well-written, and includes appropriate details explaining the methods.

3. Experiments included both in- and out-of-distribution testing to demonstrate generalizability.

**Weaknesses:**

1. While the explicit fusion of the Unet with MRF neural net model is certainly new, the mathematically groundwork for the model was all previously presented in Brudfors et al., IPMI 2019 (see Sec 2), which focused on using the MRF network alone for postprocessing segmentations.

2. The results compare the proposed MRF-Unet against vanilla Unet. I would have liked to see comparison to Unet + MRF post-processing as previously proposed in Brudfors et al., IPMI 2019, since this work is closely related.

3. Given the small number of subjects, it would have been nice if the in-distribution experiments were done in a cross-validation fashion rather than single split.

4. While for low j (number of parameters) it appears that the proposed MRF-Unet may really improve segmentation, for j>=3 it is unclear based on Fig. 4 whether the model is really better, as there is very large overlap between Unet and MRF-Unet results. Would statistical testing show that there are significant differences in segmentation performance?

5. The authors do a nice experiment where they show the effect of increasing number of model parameters on segmentation results. It seems that as the number of parameters increases, the vanilla Unet begins to meet the performance of the MRF prior model. I am wondering what the plot would look like if j=6 datapoint were included? And while the authors note the MRF model would allow lighter weight models to be used with better performance compared to Unet, the performance is increasing with increasing j, so the heavier weight models, which seem to convergence in performance, would still likely be preferred for their improved accuracy. Given this, is the MRF-Unet model as practically useful?

6. In the intro the authors state a potential benefit of MRF compared to CRF is that MRF does not depend on image data, and thus may be less prone to overfitting. It would have been nice to see comparisons to CRF models in the experiments to test this hypothesis.

**Deanonymize Review:**

no

**Detailed Comments:**

The authors mention that they use Adam optimizer with "a dynamic learning rate based on model loss" - are they just referring to the inherent scaling in Adam algorithm, or are they modifying the learning rate in some other specific way? Please clarify, and for completeness specify the Adam parameters used.

**Final Rating Justification:**

The authors reasonably addressed my questions/concerns, and I appreciate the added improvements to the paper, including the statistical testing to demonstrate significant differences and effect of using a larger model. Although I still think the math development to neural net translation highly overlaps with prior work, and I would have really liked to see a comparison to a Unet + separate MRF network used as postprocessing step as was introduced previously, I do think the presented fusion of classical estimation within a deep learning framework would be of great interest to readership. Thus, I upgrade my rating to weak accept.

**Justification Of The Preliminary Rating:**

This paper is interesting and nicely incorporates the MRF prior smoothness model on segmentation labels into an end-to-end deep learning framework.
However, as a methodology paper, I am not entirely convinced of the novelty compared to the prior work, explained above.
The experiments are promising, but some comparisons/analysis seems missing, and in the end make me wonder whether the MRF-Unet model would be useful in practice

**Paper Type:**

methodological development

**Questions To Address In The Rebuttal:**

Please see in particular points 1, 2, 4, and 5 in the "Weaknesses" section

**Special Issue:**

no

---

> ### Author Response · Authors · 2021-03-17
> **Response to AnonReviewer3**
>
> Thank you for your very comprehensive feedback, below are our responses.
>
> 1. Q: While the explicit fusion of the Unet with MRF neural net model is certainly new, the mathematically groundwork for the model was all previously presented in Brudfors et al., IPMI 2019 (see Sec 2), which focused on using the MRF network alone for postprocessing segmentations.
>
> We would like to clarify that the theory is different. In Brudfors-2019, variational bayes (VB) was used to approximate the segmentation posterior. In this paper, VB could no be used as the UNet outputs a conditional distribution over segmentation labels (i.e., not a likelihood); we therefore propose to use a products of experts model to justify our mean-field iterative approach. However, we acknowledge that there are a lot of commonalities.
>
> 2. Q: The results compare the proposed MRF-Unet against vanilla Unet. I would have liked to see comparison to Unet + MRF post-processing as previously proposed in Brudfors et al., IPMI 2019, since this work is closely related.
>
> We agree with the reviewer that comparing to Brudfors-2019 would have been a good idea, but was left out in this paper as we wanted to have space for the convergence analysis. We will however add this experiment to any future work we do.
>
> 3. Q: Given the small number of subjects, it would have been nice if the in-distribution experiments were done in a cross-validation fashion rather than single split.
>
> Thank you for this comment, it is a great point and something we actually considered doing. However, in the end we decided to stick to a traditional train/test split, as fitting the models to multiple partitions of the data would have been too time-consuming.
>
> 4. Q: While for low j (number of parameters) it appears that the proposed MRF-Unet may really improve segmentation, for j>=3 it is unclear based on Fig. 4 whether the model is really better, as there is very large overlap between Unet and MRF-Unet results. Would statistical testing show that there are significant differences in segmentation performance?
>
> Thank you for pointing this out, we have added Wilcoxon tests for statistical significance to our revised version.
>
> 5. Q: The authors do a nice experiment where they show the effect of increasing number of model parameters on segmentation results. It seems that as the number of parameters increases, the vanilla Unet begins to meet the performance of the MRF prior model. I am wondering what the plot would look like if j=6 datapoint were included? ... Given this, is the MRF-Unet model as practically useful?
>
> Thank you for this comment. Based on your point (and AnonReviewer1) we have added an additional, larger architecture to our validation (j=6), in the revised version of our submission. This showed less difference between the two methods, but did not show that adding the MRF was harmful. The practical use of our method would be when light-weight architectures are needed; this could for example be on devices outside the domain of powerful workstations or GPU clusters, or when the network has a large and complex architecture where the CNN segmentation algorithm would be just a subset of the complete model.
>
> 6. Q: In the intro the authors state a potential benefit of MRF compared to CRF is that MRF does not depend on image data, and thus may be less prone to overfitting. It would have been nice to see comparisons to CRF models in the experiments to test this hypothesis.
>
> This is definitely a fair request. The reason that we did not compare to a CRF based model is that they have been shown to not translate well to medical imaging data:
>
> Monteiro, Miguel, Mário AT Figueiredo, and Arlindo L. Oliveira. "Conditional random fields as recurrent neural networks for 3d medical imaging segmentation." arXiv preprint arXiv:1807.07464 (2018).
>
> We have added this to the revised paper.
>
> 7. Q: The authors mention that they use Adam optimizer with "a dynamic learning rate based on model loss" - are they just referring to the inherent scaling in Adam algorithm, or are they modifying the learning rate in some other specific way? Please clarify, and for completeness specify the Adam parameters used.
>
> We apologise for the unclarity. We used an approach in where the learning rate was changed based on the validation loss, where if the loss did not change much we made the learning rate smaller, and vice versa. We have clarified this in our revised version.
>
> 8. Q: The experiments are promising, but some comparisons/analysis seems missing, and in the end make me wonder whether the MRF-Unet model would be useful in practice
>
> We agree with the reviewer that the experiments could have been more exhaustive and we highly appreciate their suggestions. Still, we argue that the theoretical contribution -- modelling a likelihood+prior type model in a CNN Context -- on top of the validation results, could find an interesting readership.

---

### Official Review · AnonReviewer1 · 2021-03-07

**Confidence:** 4
**Preliminary Rating:** 3
**Recommendation:** Poster

**Summary:**

The paper proposes the use of a "product of experts" approach to combine a U-Net for semantic segmentation with a MRF prior to counter the poorer segmentation accuracy on out-of-distribution test examples. The authors present the approach and method to implement this, and present results of experiments where the approach is applied to 3D neuroimaging data (multi-class segmentation of brain tissues). They report in their experiments how the MRF-U-Net outperforms the baseline U-Net measured by the Dice coefficient.

**Strengths:**

The presentation is clear, accurate, and concise, without unnecessary sidetracks. It is sound in its methodology and comprehensive in the presentation of theoretical and practical parts, presents clear figures to underscore the approach and results, and puts the own work into context both in the introduction and the summary sections.
The combination of CNN and MRF is motivated from a different angle than previous publications (Li, CVPR 2016, Liu, TPAMI 2018) and thoroughly set up. The MRF, judged based on the presented results, both quantitatively and qualitatively provides  improved segmentations, visually apparent in the locally smoother appearance of the segmentation masks and reflected in the Dice plots.


**Weaknesses:**

* As the authors state, the inductive bias of a CNN naturally is locality, and it is not only theoretically able to capture this information, but will most certainly do so with enough training data. It is this last point, where they argue that in limited-data scenarios, a MRF could play a role, where I would have hoped for more insights. Instead of conducting a convergence analysis of the MRF iterations required, I would have liked to see a convergence analysis of from which amount of data upwards the MRF doesn't contribute any more quality to the U-Net.
* In fact, while the authors claim that the MRF-enhanced U-Net "consistently outperforms" the plain U-Net, there is not much of a difference visible anymore in U-Nets with higher parameter numbers, and it can from the plots also not be argued that a MRF-U-Net with fewer parameters should be used -- in the contrary, it keeps improving but is being approached by the conventional U-Net, so that the further improvement is likely no longer due to the MRF. Will it perhaps even deteriorate the U-Net quality if added to a U-Net with even more parameters, e.g. in a setting with much more training data? In this case, it would even be a danger to add the otherwise not harming MRF to any U-Net.
* I'm always concerned when papers compare their novel approach to a baseline that is obtained by castrating their own approach -- this essentially only proves non-inferiority. Here: the baseline is a U-Net that is perhaps not optimally set up or trained. It helps the paper, that the authors conduct experiments that show the performance over a set of U-Net configurations, though. And of course, there is no agreed gold standard segmentation method to compare to, and I'm far from advocating to use one choice of a published network, but it might make the work more convincing if compared for example to a nnUNet result, just to name one that is proven to yield a fairly consistent performance on a variety of segmentation tasks.

**Deanonymize Review:**

no

**Detailed Comments:**

In Figure 2, you show the masks for U-Net, MRF-U-Net without applying the MRF, and MRF-U-Net. How can it be that the MRF-U-Net without MRF use is so markedly different from the "solitary" U-Net?

**Justification Of The Preliminary Rating:**

The paper is concise and interesting enough to spawn discussions and inspire implementations and further research. Even without substantial changes, the soundness of the presentation make it almost acceptable to me, though I would really hope that the authors at least add preliminary experiments to tackle the questions raised.

**Paper Type:**

both

**Questions To Address In The Rebuttal:**

I think this should be clear from the "weaknesses" and the "detailed comments". I would not expect to tackle all points, but learning how the approaches with/without MRF compare on a task with more training data, and with a more powerful U-Net, would make a great leap for me.

**Special Issue:**

no

---

> ### Author Response · Authors · 2021-03-17
> **Response to AnonReviewer1**
>
> We thank the reviewer for their very insightful and constructive comments and feedback. Our responses are as follow:
>
> 1. Q: As the authors state, the inductive bias of a CNN naturally is locality, and it is not only theoretically able to capture this information, but will most certainly do so with enough training data. It is this last point, where they argue that in limited-data scenarios, a MRF could play a role, where I would have hoped for more insights. Instead of conducting a convergence analysis of the MRF iterations required, I would have liked to see a convergence analysis of from which amount of data upwards the MRF doesn't contribute any more quality to the U-Net.
>
> Thank you for this idea, it indeed would have been an interesting experiment and one we would like to implement in future work. However, we argue that the convergence analysis is important, too, as it verifies that the theory we propose resembles a monotonically increasing maximisation based on KL divergence.
>
> 2. Q: In fact, while the authors claim that the MRF-enhanced U-Net "consistently outperforms" the plain U-Net, there is not much of a difference visible anymore in U-Nets with higher parameter numbers, and it can from the plots also not be argued that a MRF-U-Net with fewer parameters should be used -- in the contrary, it keeps improving but is being approached by the conventional U-Net, so that the further improvement is likely no longer due to the MRF. Will it perhaps even deteriorate the U-Net quality if added to a U-Net with even more parameters, e.g. in a setting with much more training data? In this case, it would even be a danger to add the otherwise not harming MRF to any U-Net.
>
> Interesting observations and suggestions, thank you. Based on your points (and AnonReviewer3) we have added an additional, larger architecture to our validation (j=6), in the revised version of our submission. This showed less difference between the two methods, but did not show that adding the MRF was harmful.
>
> 3. Q: I'm always concerned when papers compare their novel approach to a baseline that is obtained by castrating their own approach -- this essentially only proves non-inferiority. Here: the baseline is a U-Net that is perhaps not optimally set up or trained. It helps the paper, that the authors conduct experiments that show the performance over a set of U-Net configurations, though. And of course, there is no agreed gold standard segmentation method to compare to, and I'm far from advocating to use one choice of a published network, but it might make the work more convincing if compared for example to a nnUNet result, just to name one that is proven to yield a fairly consistent performance on a variety of segmentation tasks.
>
> To clarify, our experiment first defined a baseline UNet model, with hyper-parameters that seemed to work well. We then only added the ‘MRF module’ to this UNet, without changing any of its hyper-parameters. We argue that using exactly the same backbone allows for an unbiased sensitivity analyses, and that pure performances can only really be assessed in the context of challenges. To the best of our knowledge, the nnUNet uses an automated method for setting the hyper-parameters, this is not what we wanted for our validation, as this would have biased our analysis.
>
> 4. Q: In Figure 2, you show the masks for U-Net, MRF-U-Net without applying the MRF, and MRF-U-Net. How can it be that the MRF-U-Net without MRF use is so markedly different from the "solitary" U-Net?
>
> Great question! A possible reason that the MRF-UNet, without any MRF iterations, looks different from the baseline UNet could be that training a combined MRF-UNet allows the UNet part to focus on certain image features, as the MRF models some of the spatial regularity. So that, when one does a forward pass though a trained MRF-UNet, without applying any mean-field iterations, some of the spatial smoothness is missing. This was the idea we had for showing examples with and without MRF iterations. We have clarified this in the Figure 2 caption.

---

### Official Review · ~Matthan_W._A._Caan1 · 2021-03-08

**Confidence:** 5
**Preliminary Rating:** 3
**Recommendation:** Oral, Poster

**Summary:**

This paper introduces a MRF prior to a classical U-Net, a MRF-UNet product, to improve the generalization to unseen data by leveraging the good performance of U-Net for image segmentation and the invariance of MRF to image data. The fused distribution of the product is approximated by a mean-field approach, allowing for back-propagation. Varying sizes of UNet architecture are evaluated for in- and out-of-distribution datasets.

**Strengths:**

The outputs of the UNet are fused with the MRF component, which then iteratively estimates the responsibility map and minimizes the cross entropy between the ground truth and the responsibility map. The network is memory and runtime efficient, thus appropriate for certain clinical applications. The paper is well structured with appropriate language, and prior work is addressed decently.

**Weaknesses:**

With larger UNet architecture (j=4,5), the difference of performance between MRF-UNet and UNet gets less significant. Can the authors elaborate on the improvement for the largest architecture? Could this difference be attenuated by using a larger dataset, considering that the current training set is relatively small?

While the paper makes the clear case of out-of-distribution performance, the shown performance will likely be inferior to conventional non-DL methods (SPM/FSL/Freesurfer). Please discuss.


**Deanonymize Review:**

yes

**Detailed Comments:**

Have the authors tried to incorporate existing techniques such as data augmentation and compare the difference between MRF-UNet and UNet again?
How would the authors summarize the “in- and out-of-distribution” of the two datasets used? More specifically, how do the (distribution of the) two datasets differ from each other? Please make this more explicit.


**Justification Of The Preliminary Rating:**

The paper addresses the question of robustly processing unseen data and the improvement is clearly shown. Still, the performance is inferior to established methods, which slighty limits the impact of the work.

**Paper Type:**

methodological development

**Questions To Address In The Rebuttal:**

Fig. 3 plots between-subject variation over within-subject improvement, making it appear as minor, not in this paper's benefit. Consider adjusting this plot.

The still imperfect segmentation on out-of-distribution data slightly limit the impact of this otherwise nice paper.

**Special Issue:**

no

---

> ### Author Response · Authors · 2021-03-17
> **Response to AnonReviewer4**
>
> We thank the reviewer for their feedback and are pleased to hear that they enjoyed reading the paper. Below, we hope that we address your questions and concerns:
>
> 1. Q: With larger UNet architecture (j=4,5), the difference of performance between MRF-UNet and UNet gets less significant. Can the authors elaborate on the improvement for the largest architecture? Could this difference be attenuated by using a larger dataset, considering that the current training set is relatively small?
>
> We agree with the reviewer that increasing the subject numbers could make the difference more noticeable, for larger architectures. Another explanation could be that the ‘MRF module’ is more useful for lower parameter models, which produces ‘noisier’ decoder outputs.
>
> 2. Q: While the paper makes the clear case of out-of-distribution performance, the shown performance will likely be inferior to conventional non-DL methods (SPM/FSL/Freesurfer).
>
> Our approach is yet not on par with probabilistic techniques (often mixture model based) for out-of-distribution data. However, readers familiar with the non-DL methods mentioned might find the connection we show interesting: the likelihood+prior separation. Additionally, there are CNN based methods that uses templates, which mimics some probabilistic techniques and has shown to generalise quite well, e.g:
>
> Dalca, Adrian V., et al. "Unsupervised deep learning for Bayesian brain MRI segmentation." International Conference on Medical Image Computing and Computer-Assisted Intervention. Springer, Cham, 2019.
>
> Our approach could easily be plugged into this type of network.
>
> 3. Q: Have the authors tried to incorporate existing techniques such as data augmentation and compare the difference between MRF-UNet and UNet again? How would the authors summarize the “in- and out-of-distribution” of the two datasets used? More specifically, how do the (distribution of the) two datasets differ from each other? Please make this more explicit.
>
> Thank you for your comment. The implementation section of our paper specified that we used various augmentation techniques: nonlinear deformation, intensity-non uniformities and additive noise; to improve the fit on the small datasets. The Data section additionally explains the differences between the two datasets: different centres+scanners, pathology vs no pathology, age distribution. To clarify we have added a reference to Figure 2, which shows examples of the two datasets, to the Data section.
>
> 4. Q: Fig. 3 plots between-subject variation over within-subject improvement, making it appear as minor, not in this paper's benefit. Consider adjusting this plot.
>
> Thank you for this comment. We agree that plotting the Dice scores in their ‘raw’ form (e.g. without any type of normalisation) could have the effect suggested; however, we here prefer showing what dice and variance to expect in a ‘practical situation’.
>
> 5. Q: The still imperfect segmentation on out-of-distribution data slightly limit the impact of this otherwise nice paper.
>
> Thank you. Although our results are not perfect, we aimed to take a step in the direction towards segmenting out-of-distribution data. We hope that the experiment in Figure 3 supports this.
>
> 6. Q: Still, the performance is inferior to established methods, which slighty limits the impact of the work.
>
> We would like to point out that our approach can be easily plugged into any type of CNN technique, for example, such models that has been shown to perform well on out-of-distribution data. Where doing so could improve their results even further. We argue this is where some of the paper’s impact lays.

---

### Official Review · AnonReviewer2 · 2021-03-08

**Confidence:** 4
**Preliminary Rating:** 3

**Summary:**

Use of Markov random fields (MRFs) over labels and Unet CNN.
Propose to fuse both strategies by computing the product of distributions of a UNet and a MRF.
Solved as an approximate distribution using an iterative mean-field approach.
Resulting MRF-UNet is trained jointly by back-propagation.
Applied on 3D neuroimaging data for generalisation to out-of-distribution samples.
Discusses low computational burden of MRF component.


**Strengths:**

Paper is very similar (especially all maths) to paper from same 1st author in Brudfors-2019.  But clear novelty claimed to now use the MRF-NN model with a UNet instead of a  probabilistic segmentation model.

Some efforts to clearly explain the VB process and introduction of the final architecture (Fig 1, Algo 1) .
Training implementation uses interesting tricks (e.g. randomly sample the number of iterations from a discrete uniform distribution during training).




**Weaknesses:**

Unfortunately, visual results are not very convincing. The smoothing/regularization effect which can be easily obtained on tissue segmentation via classic Markovian regularization is not obtained here.

"Segmenting out-of-distribution images is clearly a very challenging task": problem does not appear to be solved here.

Why training on such a small data set? Why using the same exact cohorts as Brudfors-2019 and not comparing your results to this paper?

Fig3: Dice scores should be shown per tissue class.


**Deanonymize Review:**

no

**Justification Of The Preliminary Rating:**

This paper is very pleasant to read regarding the methodological design and its mathematics.
But the end of the story is challenging to follow (more details on how to link Algo 1 and the provided Equations?)
And overall, results are not very strong (or maybe I missed something for Fig.2 MRI(out) and Dice <0.7 in Fig 3)

**Paper Type:**

both

**Special Issue:**

no

---

> ### Author Response · Authors · 2021-03-17
> **Respone to AnonReviewer2**
>
> Thank you for your time, and both valuable and detailed comments. We are glad to hear you enjoyed reading our submission. Below we have attempted to address your concerns:
>
> 1. Q: Paper is very similar (especially all maths) to paper from same 1st author in Brudfors-2019. But clear novelty claimed to now use the MRF-NN model with a UNet instead of a probabilistic segmentation model.
>
> We would like to clarify that the maths is different. In Brudfors-2019, variational bayes (VB) was used to approximate the segmentation posterior. In this paper, VB could no be used as the UNet outputs a conditional distribution over segmentation labels (i.e., not a likelihood); we therefore proposed using a products of experts model to justify our mean-field iterative approach. However, we acknowledge that there are a lot of commonalities.
>
> 2. Q: Unfortunately, visual results are not very convincing. The smoothing/regularization effect which can be easily obtained on tissue segmentation via classic Markovian regularization is not obtained here.
>
> We would like to point out that our MRF has non-linearities (and use slightly higher order cliques than usual -- 26 neighbours instead of 6 in many classic schemes). We could therefore expect more than ‘just’ smoothing (the main point of the method is that we are learning patterns at the label levels, and that these patterns should be transferable across intensity profiles, so a more ‘complex’ MRF encoding should be beneficial). Additionally, we argue that Figure 2 does show some of the typical MRF smoothing effect; albeit, perhaps, not very strongly. Additionally, a stronger, visual, smoothing effect does not necessarily imply a better model fit.
>
> 3. Q: "Segmenting out-of-distribution images is clearly a very challenging task": problem does not appear to be solved here.
>
> We think that our method is a step in that direction, but as the reviewer points out, not a solution. Our intention was to tackle this problem by separating label patterns from image features. We argue that the experiment in Figure 3 shows that the MRF-UNet does improve Dice scores at this task, compared to a baseline approach.
>
> 4. Q: Why training on such a small data set? Why using the same exact cohorts as Brudfors-2019 and not comparing your results to this paper?
>
> We again used those two datasets as they contain the same manual brain labels, which facilitated our in- and out-of-distribution comparison, and because there exist a limited amount of such publicly available data (at least to our knowledge). We agree with the reviewer that comparing to Brudfors-2019 would have been a good idea, but was left out in this paper as we wanted to have space for the convergence analysis. We will however consider this addition to any subsequent validation of our method.
>
> 5. Q: This paper is very pleasant to read regarding the methodological design and its mathematics. But the end of the story is challenging to follow (more details on how to link Algo 1 and the provided Equations?) And overall, results are not very strong (or maybe I missed something for Fig.2 MRI(out) and Dice <0.7 in Fig 3)
>
> Thank you. We agree that the theory is not explained self-enclosed in the paper, to a reader not familiar with the type of approximate inference techniques we use; but we do provide references to sources we think explain them well. In an attempt to clarify, we have added a reference to the corresponding update equation in the algorithm box. Regarding the absolute Dice scores obtained: we did not aim to maximise dice scores, but to show that our proposed approach improved on a baseline method. We argue that our validation shows this, and that the MRF is a very light-weight layer that can be added to any segmentation CNN and will often/always be beneficial  (at worse neutral).

---

### Author Response · Authors · 2021-03-17
**Response to all reviewers**

We would like to thank the reviewers for their insightful and detailed comments, as well as for pointing out their concerns. We have attempted to address all of these in separate responses below. We would further like to point out that we have added a link to our implementation in the revised version of the manuscript (https://github.com/balbasty/nitorch --> https://github.com/balbasty/nitorch/blob/f10d5e142652c201623d49354b878cb5c7009047/nitorch/nn/modules/segmentation.py#L394).

---

### Meta-Review · Area_Chair1 · 2021-03-24

**Recommendation:** Accept (Poster)

**Metareview:**

This paper proposes a MRF-UNet product to improve the generalization to unseen data. The reviewers pointed out the high similarity and overlap with IPMI Brudfors-2019. Moreover, they questioned  the results, which are lower than current state-of-the-art.  Consequently, the practical use of this method seems quite limited. However this paper is well-written and the tackle problem is very important.

**Paper Type:**

methodological development

---

### Decision · Program_Chairs · 2021-03-31

Accept